# Cattle-related occupational accidents in Japan

Hilomi Iwai[1]*, Hideki Yamamoto[1,2]

1 Laboratory of Environmental Health and Global Health, Faculty of Pharma-Science, Teikyo University, Itabashi city, Tokyo, Japan, 2 Health Services Research and Development Centre, Faculty of Medicine, Department of Health Services Research, University of Tsukuba, Tsukuba city, Ibaraki, Japan

These authors contributed equally to this work.
* h.iwai@pharm.teikyo-u.ac.jp

## Abstract

Cattle-related injuries are a significant contributing factor in 84.3% of livestock-related incidents in Hokkaido Japan. The specific countermeasures to respond to its characteristics are needed because cattle move freely. This study examined the JA Kyosai mutual aid payment claim form data of the agricultural work-related incidents that occurred from 1st January 2013 to 31st December 2016. We extracted incidents coded 'Cattle' term on the 'Causing Stuffs/Animals' column as cattle-related injuries and analysed them. In four years, there were 227 cattle-related incidents and all were non-fatal cases. The most common age group of victims were in their 60s (37.9%). The Therapy duration was 1 to 243 days, with one day being the most common (10.1%); however, in 51.5% of the cases it required more than 30 days. The most common Activity at the moment of the incident was 'bringing the cattle' (22.5%), of which 'leading the cattle by rope' (56.9% of bringing) was most frequently mentioned. Even farmers with significant experience with cattle cannot control cattle proficiently. Neither can they predict cattle behaviour. Hence, cattle should be kept under protected contact. Since prevention of incidents has limits, harm reduction to farmers must be considered. In the short-term planning, isolated pathways and shock-absorbing lead rope are effective in preventing injuries. However, a more long-term perspective must consider, a fully automated system that reduces human contact with cattle on farms.

## Introduction

In Japan, agriculture is a hazardous industry. The number of annual agriculture-related fatal incidents is estimated at 378 cases [1]. Details of agriculture-related incidents must be analysed to develop effective measures in reducing such incidents. However, agriculture-related incidents are not included in the Survey on Industrial Incidents, of the Ministry of Health, Labour and Welfare of the Japanese government as 98% of Japan's agricultural management entities are family-owned (self-employed) [2] and have no compulsory occupational incident insurance or obligation to report occupational incidents. According to the national population census of Japan in 2015, 2 million people are working for agriculture; not only them but many people are working for seasonal part-time job or family helper.

**Data Availability Statement:** The data underlying the results presented in the study are available from JA Kyosai (https://www.ja-kyosai.or.jp/).

**Funding:** The authors received no specific funding for this work.

**Competing interests:** No. The authors have declared that no competing interests exist.

Some prefectures conduct their own surveys of agriculture-related incidents, but their quality is inconsistent. According to the research of Zenkyoren (National Mutual Insurance Federation of Agricultural Cooperatives), large agricultural machinery was responsible for many fatal incidents and safety equipment on machinery was suggested to reduce the incidence [3]. However, specific countermeasures are needed to respond to the characteristics of these incidents since safety equipment cannot be exist on livestock operations. Cattle are the biggest cause of livestock-related incidents in Hokkaido, Japan (84.3%) [4]. Even if cattle look gentle, they are a kind of large herbivore. Zoos that also deal with animals are required to keep large herbivore under protected contact as dangerous animals by the Ministry of the Environment of the Japanese government having jurisdiction over zoos. However, in farm, it is extremely difficult to avoid direct contact with them, particularly during milking, handling, transporting and medical treatment.

This study aims to clarify the occupational incidents related to cattle rearing and seeks measures to reduce the damages caused by these incidents.

## Method

### Data source

In this study, we analysed JA Kyosai mutual aid payment claim form data. JA Kyosai is the cooperative insurance conducted by Zenkyoren (National Mutual Insurance Federation of Agricultural Cooperatives), administered by JA (Japan Agricultural Cooperative). In Japan, JA is the only national cooperative of farmers, based on the Agricultural Cooperative Act. JA had 10.26 million members in 2015 and had 658 branches in 2016. JA Kyosai does not disclose the number of policyholders for individual insurance products, but it of comprehensive life insurance is about 20 million; this number includes ordinary consumers who are not agricultural worker.

From JA Kyosai, data regarding 19,938 agricultural practice-related injury mutual aid payment claim forms were provided. These incidents had occurred between 1st January 2013 and 31st December 2016.

### Analyses

We extracted incidents coded 'Cattle' on the 'Causing Stuffs/Animals' column as cattle-related injuries. The 'Cattle' coded on 'Causing Stuffs/Animals' does not distinguish beef or dairy cattle, and feeding procedures. According to the 'Situation report of the incident', the 'Activity' at the moment of the incident was recorded. 'Therapy duration' is the period from day 1 'Date of onset / first visit / start of treatment' to 'Date of last visit / end of treatment'.

All analysis was done using Python (3.9.7) with Spyder (5.1.5). To describe the trend of the cattle-related injuries, the numbers of each label, coded in each column, were counted by pandas and graphs were drowned by seaborn.

### Ethical consideration

This study was approved by the Teikyo University Ethical Review Board for Medical and Health Research Involving Human Subjects (No. 20–221). The Review Board decided that obtaining informed consent from each individual is not required because this study uses secondary data and personally identifiable information in the data provided is blinded. Instead, a public announcement was carried out by the Review Board and JA.

## Results

### Demographics

Between 1st January 2013 and 31st December 2016, 227 cattle-related injuries had occurred, with no fatality cases. Of these, 149 victims included males (65.6%) and the average age of the victims was 58.19 years (95% CI 23.5–81); the youngest was eight years old while the eldest was 82 years old. Maximum injuries occurred in the age group of 60–69 years (86 cases, 37.9%), followed by the age groups 50–59 years (48 cases, 21.1%) and then 70–79 years (33 cases, 14.5%) (Table 1).

### Burden of the injuries

The number of cases that required hospital admission were 48 (21.1%), those that did not require admission in hospitals were 178 (78.4%), and there was one unknown (0.4%) case. The duration of therapy required was between 1 and 243 days, with minimum therapy duration being one day (23 cases, 10.2%) (Table 2).

Longer therapy was more frequent among older victims (Fig 1).

### Mechanism

The most common mechanism was 'Pulled' (39 cases, 17.2%), followed by 'Trampled' (37 cases, 16.3%) and 'Kicked' (36 cases, 15.9%) (Table 3). Therapy duration was higher density in shorter days at every mechanism (Fig 2).

### Activity at the moment of incident

The most common activity at the moment of the incidents was 'bringing' (51 cases, 22.5%), followed by 'cleaning ' (21 cases, 9.3%) and 'milking ' (21 cases, 9.3%) (Table 4). Therapy duration was more fleacent in shorter days at every activity at the moment of incident (Fig 3). With regards to 'bringing', the most common reason was 'leading the cattle by rope' (29 cases, 56.9%) (Table 5). The Therapy duration of leading the cattle by rope was between 1 and 198 days. In two cases, longer therapy was required while other cases required 1 to 68 days of therapy (Fig 4).

**Table 1. Demographics.**

|  |  | N = 227 | |
|---|---|---|---|
|  |  | **Number** | **%** |
| Sex | Male | 149 | 65.6 |
|  | Female | 78 | 34.4 |
| Age (years) | Average | 58.19 (23.5–81) | |
|  | Minimum | 8 | |
|  | Maximum | 82 | |
| Age groups | Under 30 | 10 | 4.4 |
|  | 30–39 | 21 | 9.3 |
|  | 40–49 | 20 | 8.8 |
|  | 50–59 | 48 | 21.1 |
|  | 60–69 | 86 | 37.9 |
|  | 70–79 | 33 | 14.5 |
|  | Above 80 | 9 | 4.0 |

**Table 2. Burden of the injuries.**

|  |  | Number (N = 227) | % |
|---|---|---:|---:|
| Fatality (Number) | Fatal | 0 | 0.0 |
|  | Non-fatal | 227 | 100.0 |
| Admission (Number) | Admitted | 48 | 21.1 |
|  | Non-admitted | 178 | 78.4 |
|  | Unknown | 1 | 0.4 |
| Therapy duration (Number of days) | 1 | 23 | 10.1 |
|  | 2–4 | 1 | 0.4 |
|  | 5–9 | 29 | 12.8 |
|  | 10–19 | 33 | 14.5 |
|  | 20–29 | 24 | 10.6 |
|  | 30–59 | 62 | 27.3 |
|  | 60–119 | 37 | 16.3 |
|  | 120–243 | 18 | 7.9 |

## Human body region of injury

The most common body region of injury was lower extremity (50 cases, 22.0%) followed by upper extremity (41 cases, 18.1%) and abdomen/chest (34 cases, 15.0%) (Table 6). Therapy duration was higher density in shorter days at every body region of injury (Fig 5).

## Symptoms

The most common symptom among injured livestock workers was fracture or dislocation (105 cases, 46.3%), followed by bruises/abrasions/contusions/distortions (65 cases, 28.6%) and lacerations/incision wounds/crush injuries (23 cases, 10.1%) (Table 7). Therapy duration was higher density in shorter days except for fracture/dislocation. Therapy duration at fracture/dislocation peaked at longer days (Fig 6).

## Discussion

This is the first study that analyse of cattle-related injuries and fatality in Japan using the JA Kyosai mutual aid payment clam form data. In the four years between January 2013 and December 2016, 227 cattle-related injuries were recorded but all were non-fatal cases. The most common age group of victims was in their 60s and the therapy duration required 1 to 243 days, with more than 50% of the cases requiring over 30 days to recover. In Japan, the therapy duration is a severe burden on farms because the number of workers on each farm is few and it heavily depends on family members. In 2009, the average manpower and average family member manpower were 3.1 and 2.9 in dairy cattle farming in Hokkaido, 2.8 and 2.5 in dairy cattle farming in other prefectures and 1.7 and 1.4 in beef cattle farming, respectively [5].

The most common activity at the moment of the incident was bringing (51 cases, 22.5%) of which 56.9% (29 cases) was due to leading by rope. When a farmer leads the cattle by rope, they are connected directly, and a farmer is likely to be affected by the movement of the cattle. This is different from the other kinds of operations.

The data from JA Kyosai mutual aid payment claim form includes only claimed cases and does not include the not-claimed cases. Hence, it may not be representative of the actual incidence rates of incident-related injuries by cattle. Despite these limitations, JA Kyosai mutual

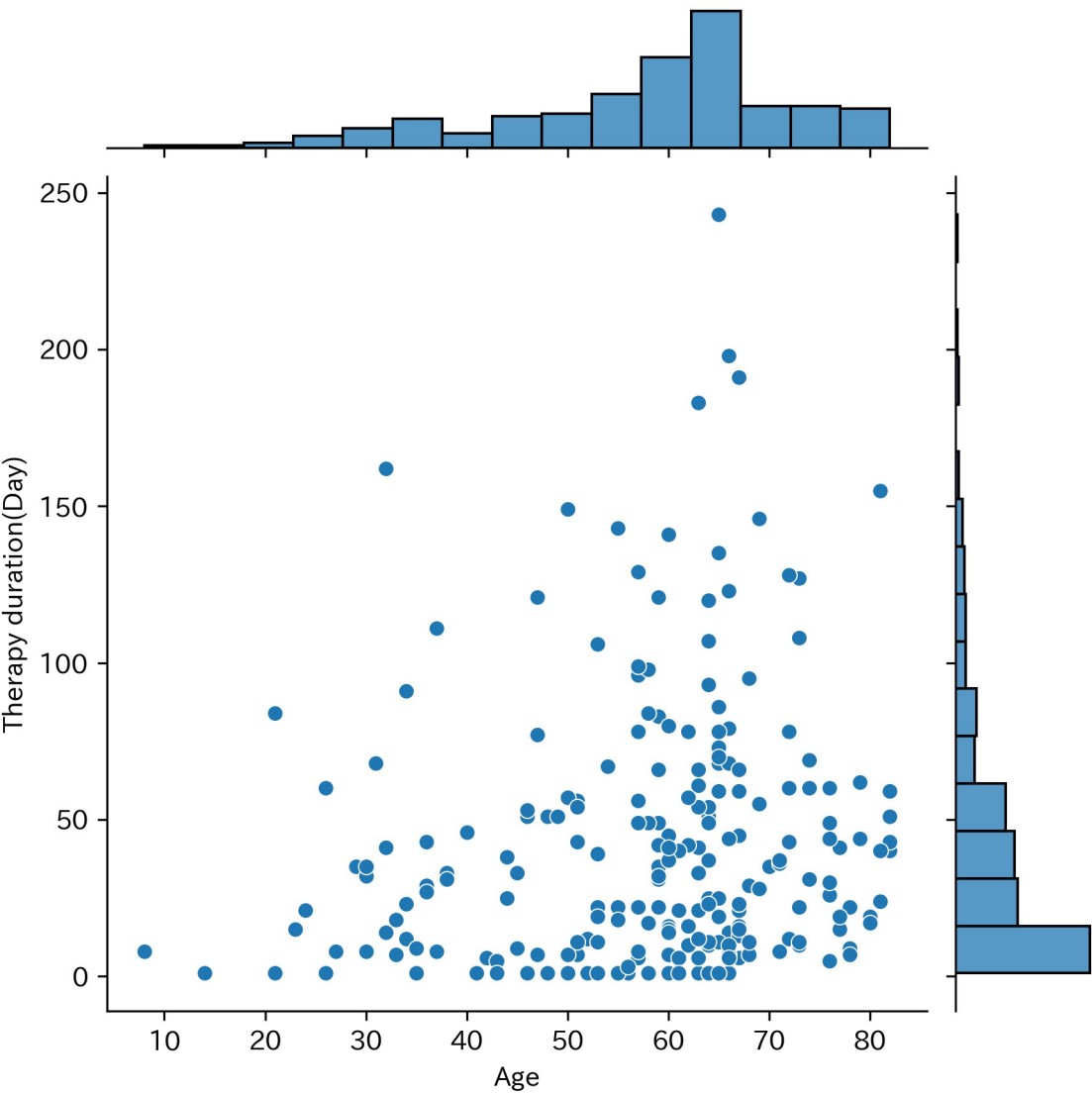

**Fig 1. Scatter plot and histogram of Therapy duration (Day) and age.**

aid payment claim form data is the biggest and only data source containing detailed information on incidents during agricultural operations in Japan. Approximately 98% of the farmers in Japan are self-employed and their farms are not included in the Survey on Industrial Incidents of the Ministry of Health, Labour and Welfare of the Japanese government. And a

**Table 3. Mechanisms of the incident.**

|  | Number (N = 227) | % |
|---|---:|---:|
| Pulled | 39 | 17.2 |
| Trampled | 37 | 16.3 |
| Kicked | 36 | 15.9 |
| Caught in cattle and construction | 26 | 11.5 |
| Other kind of clashes | 70 | 30.8 |
| Others/Unknown | 19 | 8.4 |

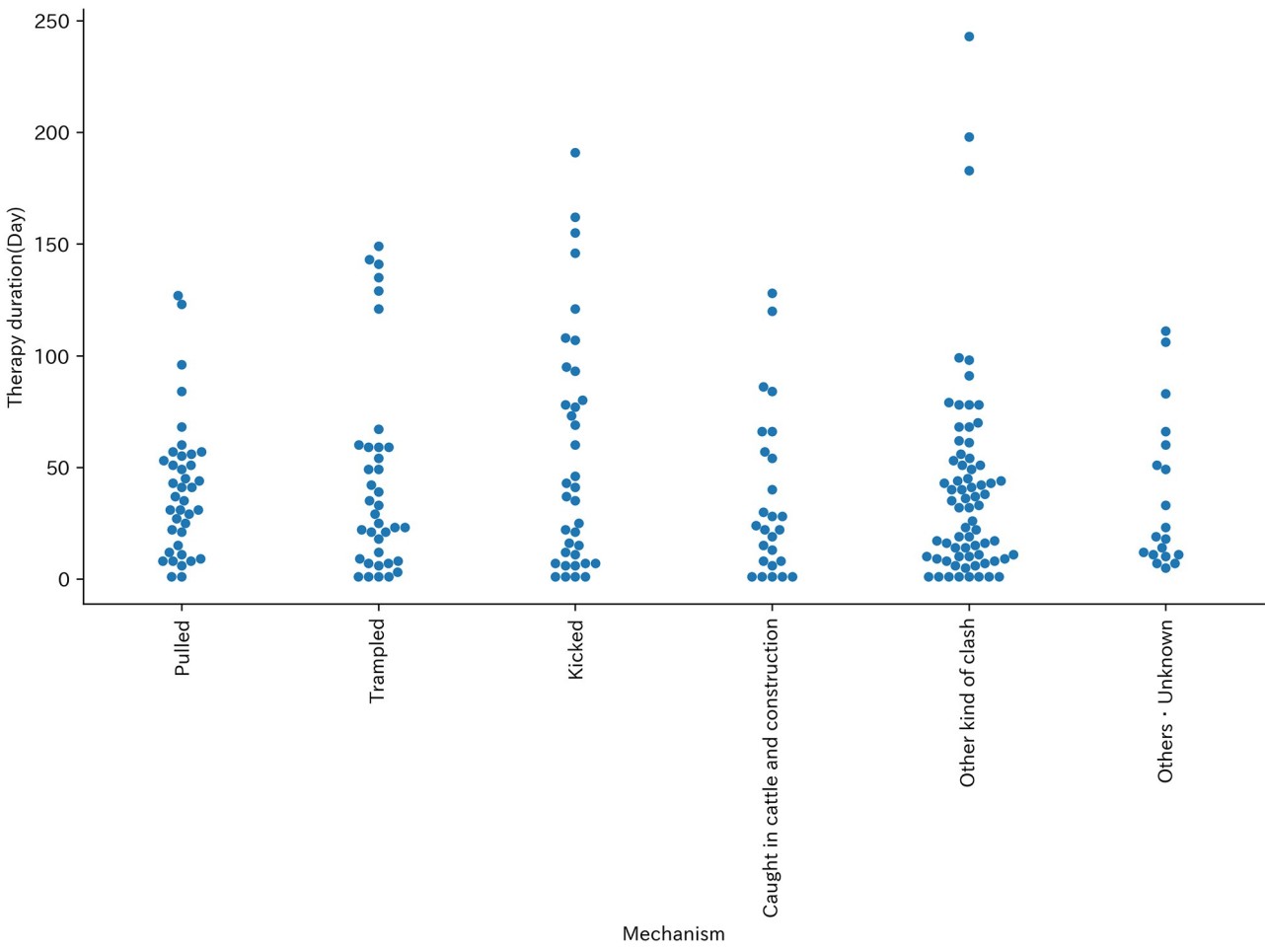

**Fig 2. Swarm plot of mechanism and Therapy duration (Day).**

**Table 4. The activity at the moment of incident.**

|  | Number (N = 227) | % |
|---|---|---|
| Bringing | 51 | 22.5 |
| Cleaning | 21 | 9.3 |
| Milking | 21 | 9.3 |
| Feeding | 14 | 6.2 |
| Taking care | 13 | 5.7 |
| Medical treatment | 10 | 4.4 |
| Shipping | 10 | 4.4 |
| Hoof trimming | 8 | 3.5 |
| Mooring | 7 | 3.1 |
| Dehorning | 5 | 2.2 |
| Assistance with delivery | 4 | 1.8 |
| Exercise | 3 | 1.3 |
| Maintenance | 3 | 1.3 |
| Others | 28 | 12.3 |
| No detail | 29 | 12.8 |

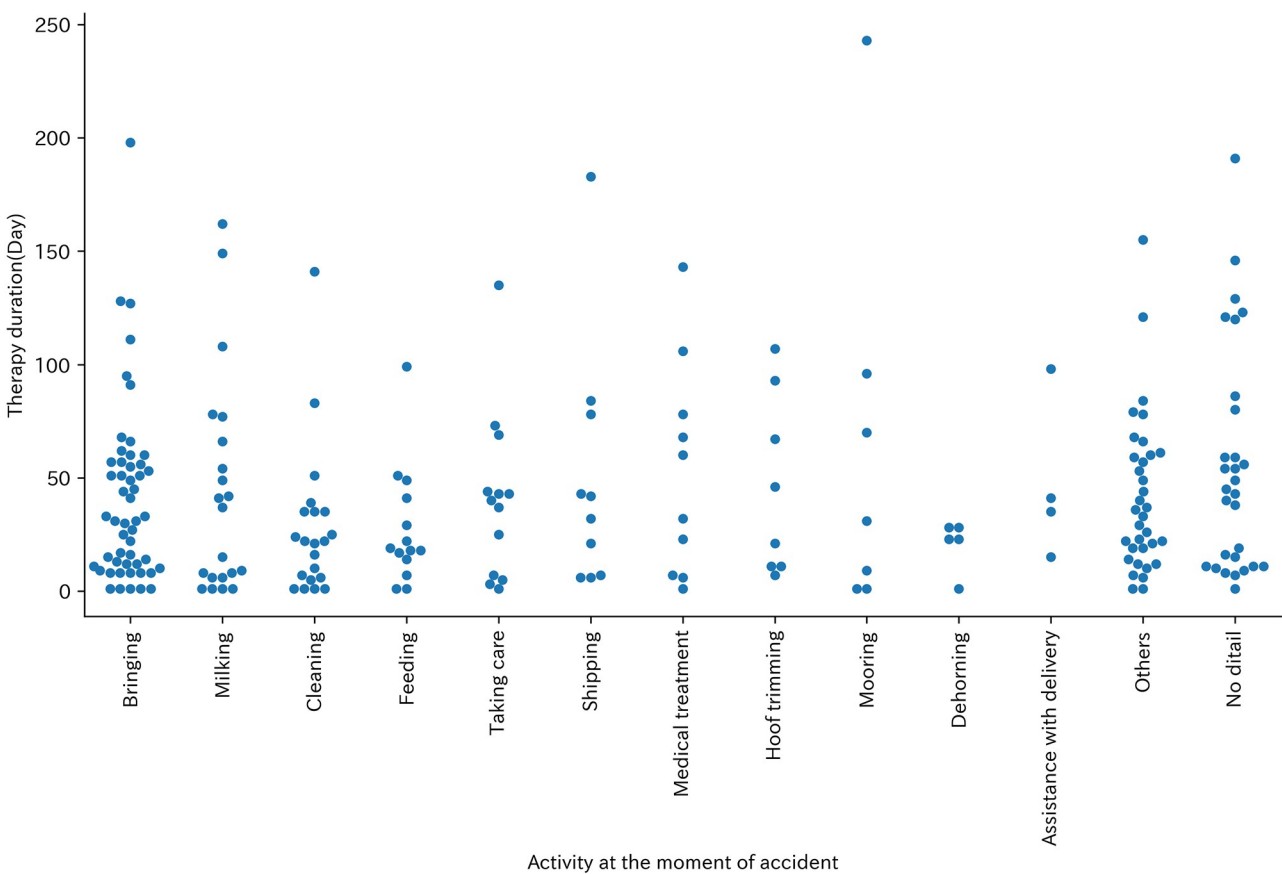

**Fig 3. Swarm plot of operation at the moment of accident and Therapy duration (Day).**

strength of the JA Kyosai mutual aid payment claim form data is that it contains detailed information of incidents in Japan as a whole.

## Conclusion

Beef and milk cattle are common farm animals. However, they are large herbivorous need to keep with caution. Even farmers with significant experience with cattle cannot predict their behaviour. Measures of harm reduction in incident should be taken at the same time as measures of prevention. For short run planning, isolated pathways and shock-absorbing leading ropes for cattle may be effective short-term measures to this end. However, in the long-term, a

**Table 5. Type of bringing.**

|  | Number (n = 51) | % |
|---|---|---|
| Leading by rope | 29 | 56.9 |
| Driving | 4 | 7.8 |
| Guiding a calf | 1 | 2.0 |
| Pushing the cattle | 1 | 2.0 |
| No detail | 16 | 31.4 |

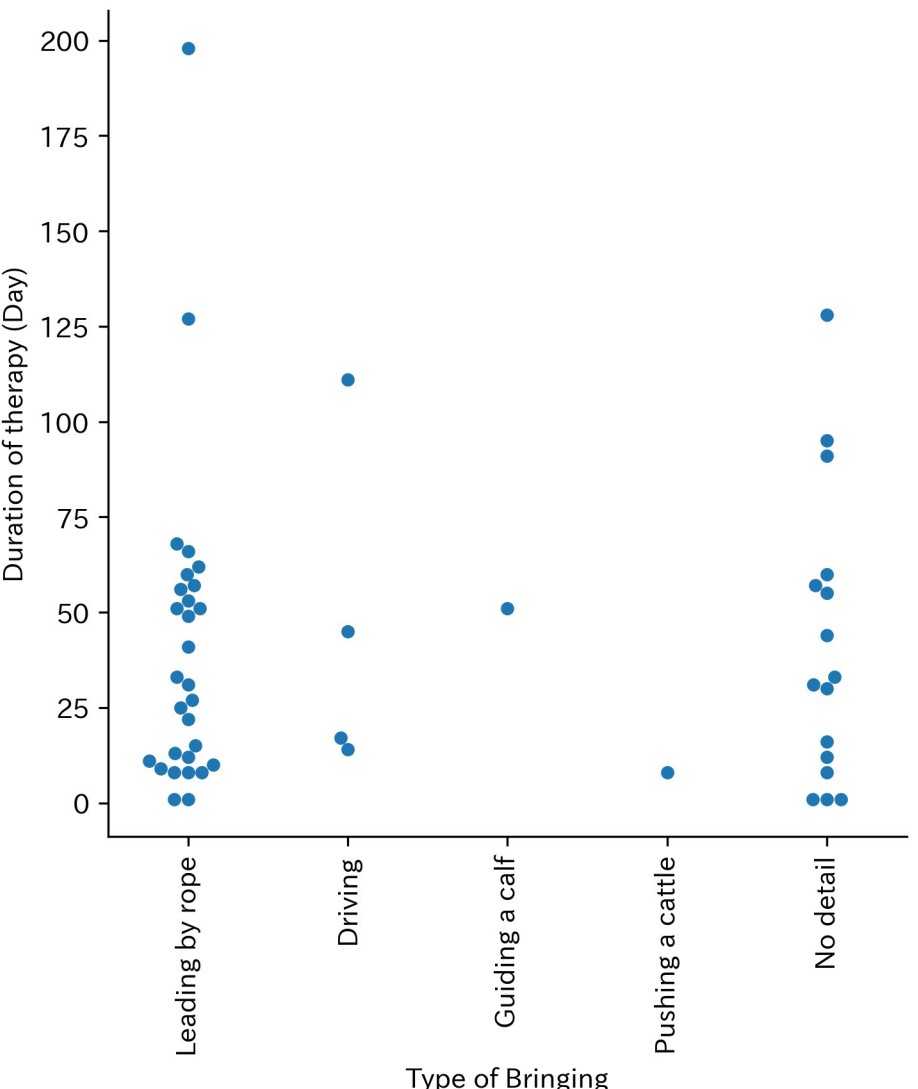

**Fig 4. Swarm plot of type of bringing and Therapy duration (Day).**

**Table 6. Human body region of injury.**

|  | Number (N = 227) | % |
|---|---|---|
| Lower extremity | 50 | 22.0 |
| Upper extremity | 41 | 18.1 |
| Abdomen/Chest | 34 | 15.0 |
| Fingers | 29 | 12.8 |
| Back/Lumbar/Buttock | 27 | 11.9 |
| Toe | 22 | 9.7 |
| Face | 12 | 5.3 |
| Head | 4 | 1.8 |
| Whole body | 3 | 1.3 |
| Globe | 2 | 0.9 |
| Neck | 2 | 0.9 |
| Teeth | 1 | 0.4 |

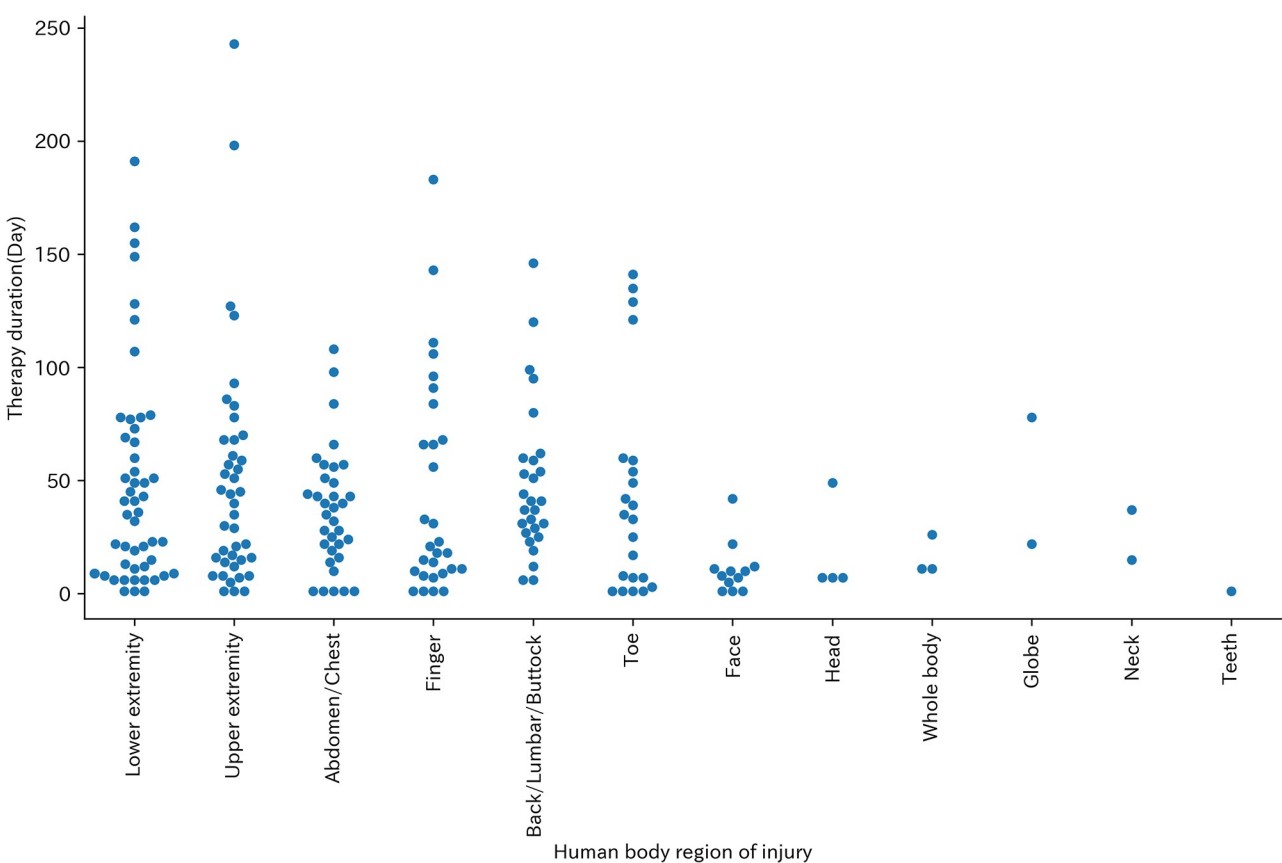

**Fig 5. Swarm plot of body region of injury and Therapy duration (Day).**

**Table 7. Symptoms.**

|  | Number (N = 227) | % |
|---|---|---|
| Fracture/Dislocation | 105 | 46.3 |
| Bruise/Abrasion/Contusion/Distortion | 65 | 28.6 |
| Laceration/Incision wound/Crush injury | 23 | 10.1 |
| Muscle/Tendon Injury | 20 | 8.8 |
| Organ injury (Surgery)/Ocular trauma | 3 | 1.3 |
| Defective/Cleavage | 2 | 0.9 |
| Haemorrhage (Intracranial, Intraocular) | 2 | 0.9 |
| Organ injury (non-surgery) | 2 | 0.9 |
| Bite wound (Animal, Snake) | 1 | 0.4 |
| Code was not entered | 1 | 0.4 |
| Insect sting | 1 | 0.4 |
| Others | 2 | 0.9 |

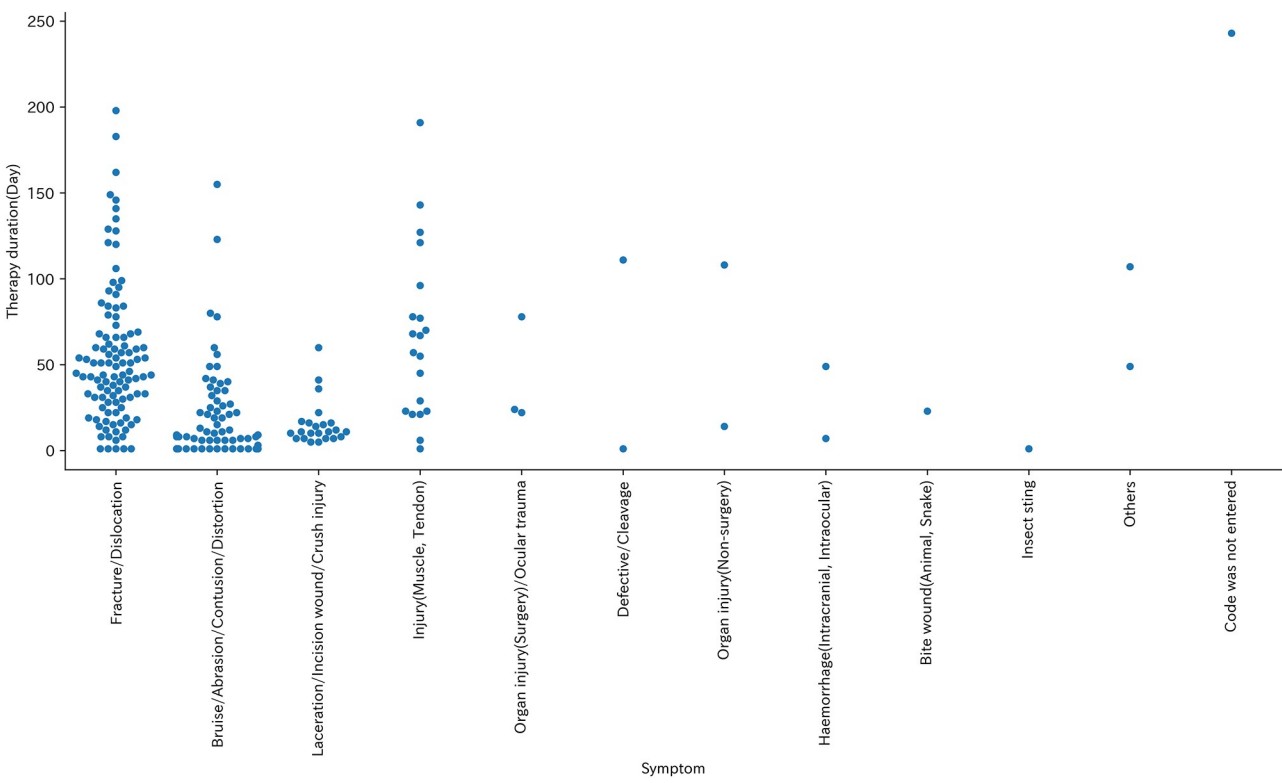

**Fig 6. Swarm plot of symptom and Therapy duration (Day).**

fully automated system which doesn't need humans for their work process should be introduced.

## Acknowledgments

We would like to thank JA Kyosai for providing JA Kyosai mutual aid payment claim form data for this study.

## Author Contributions

**Conceptualization:** Hilomi Iwai, Hideki Yamamoto.

**Data curation:** Hideki Yamamoto.

**Formal analysis:** Hilomi Iwai.

**Funding acquisition:** Hideki Yamamoto.

**Methodology:** Hilomi Iwai.

**Project administration:** Hilomi Iwai.

**Writing – original draft:** Hilomi Iwai.

**Writing – review & editing:** Hilomi Iwai, Hideki Yamamoto.

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
