## [Decision Letter · Decision Letter 0]

25 Aug 2022

PONE-D-22-16745Cattle - related injuries in JapanPLOS ONE

Dear Dr. IWAI,

Thank you for submitting your manuscript to PLOS ONE. After careful consideration, we feel that it has merit but does not fully meet PLOS ONE’s publication criteria as it currently stands. Therefore, we invite you to submit a revised version of the manuscript that addresses the points raised during the review process.

We look forward to receiving your revised manuscript.

Kind regards,

Pramod K Pandey

Academic Editor

PLOS ONE

Journal Requirements:

    "No. The funders had no role in study design, data collection and analysis, decision to publish, or preparation of the manuscript."

Additional Editor Comments:

Dear Author: This study has novel information beneficial to stakeholders. However, authors are required to follow guidelines of PLOSONE and Improve data presentation (figure & table). Please improve and write abstract and conclusions concisely describing the major findings in laymen language as well as with scientific evidence. While revising, please pay special attention on English language and edits.

Reviewers' comments:

Reviewer's Responses to Questions

**Comments to the Author**

1. Is the manuscript technically sound, and do the data support the conclusions?

Reviewer #1: Yes

Reviewer #2: Yes

2. Has the statistical analysis been performed appropriately and rigorously? 

Reviewer #1: Yes

Reviewer #2: Yes

3. Have the authors made all data underlying the findings in their manuscript fully available?

Reviewer #1: Yes

Reviewer #2: Yes

4. Is the manuscript presented in an intelligible fashion and written in standard English?

Reviewer #1: Yes

Reviewer #2: No

5. Review Comments to the Author

Reviewer #1: The manuscript present a EDA of the cattle involved occupational accidents related to and provides an insight on health and safety issues in cattle production through claim data between 2013 to 2016. However, having access to a large dataset could be used to find more complex modelling and providing a better picture in future.

Reviewer #2: All comments are attached in the PDF which I included all my corrections and feedback for the authors regarding the manuscript titled cattle-related injuries and fatalities in Japan. Great work if you consider all comments from all reviewers

6. PLOS authors have the option to publish the peer review history of their article (what does this mean?). If published, this will include your full peer review and any attached files.

Reviewer #1: No

Reviewer #2: **Yes: **Mahmoud M. Nour, Ph.D., Purdue University, USA

---

## [Author Response · Author response to Decision Letter 0]

18 Jan 2023

Dear Reviewer #1

We thank the reviewer for the positive comment. Gathering the information of agricultural occupational accidents is under consideration issues. In Japan, the most of agriculture-related accidents are not included in the Survey on Industrial Accidents (by Ministry of Health, Labour and Welfare of the Japanese government) because 98% of Japan's agricultural management entities are family-owned (self-employed) and have no compulsory occupational accident insurance or obligation to report. We expect the positive approach by the related government ministries and agencies.

Thank you.

Dear Reviewer #2 Dr Mahmoud M. Nour

Thank you very much for providing important comments. We are appreciate for the time and energy you expended. We revised as follows.

Line17 etc. (Line19 etc. in the new file): ‘Incident’ or ‘Accident’

We revised the ward according to your advice. We used ‘Accident’, because ‘Accident’ is cases which results in fatal or non-fatal injury and ‘Incident’ is cases witch results in non-human injury: analysed JA Kyosai mutual aid payment claim form data include only the treated in hospital cases. However, event is ‘incident’ in Epidemiology as your advice. Therefore, we revised ‘Accident’ to ‘Incident’.

Line56, 178, 179(Line73 to 89, 259 to 263 in the new file): zoos

 We revised the text to reflect according to your advice as follow. We wanted to explain that cattle are kind of large herbivores need to be kept with caution. But the sentences of the first edition were little bet difficult to make sense.

Line56(Line73 to 89)

Before: ‘Although cattle, a kind of large herbivore, might not look dangerous, the Ministry of the Environment of the Japanese government recommends keeping them under protected contact in zoos.’

After: ‘Even if cattle look gentle, they are a kind of large herbivore. Zoos that also deal with animals are required to keep large herbivore under protected contact as dangerous animals by the Ministry of the Environment of the Japanese government having jurisdiction over zoos.’

Line178,179(Line259 to 263 in the new file)

Before: ‘ Cattle are common farm animals; under protected contact keeping is recommended in zoos by the Ministry of the Environment of the Japanese government. Not only in zoos but also in farms, cattle should be kept under protected contact. Even farmers with long and varied experience with cattle cannot predict their behaviour. Preventions have limits and harm reduction must be considered on the other hand.’

After: ‘Beef and milk cattle are common farm animals. However, they are large herbivorous need to keep with caution. Even farmers with significant experience with cattle cannot predict their behaviour. Measures of harm reduction in incident should be taken at the same time as measures of prevention.’

Line127(Line 187 in the new file): ‘Bringing’ or ‘Handling’

 We would like to keep ‘Bringing’. Because ‘bringing’ of ‘Activity at the moment of incident’ means ‘let cattle move from some place to another place’. And ‘Handling’ sometimes means ‘general breeding work’. Therefor we chose ‘bringing’ according to the previous study (David I. Douphrate et.al. Livestock-Handling Injuries in Agriculture: An Analysis of Colorado Workers’ Compensation Data. AMERICAN JOURNAL OF INDUSTRIAL MEDICI, 2009;52:391-407.). 

Other pointed wards and sentences

 We revised according to your advice.

---

## [Decision Letter · Decision Letter 1]

12 Mar 2023

PONE-D-22-16745R1Cattle - related injuries in JapanPLOS ONE

Dear Dr. IWAI,

Thank you for submitting your manuscript to PLOS ONE. After careful consideration, we feel that it has merit but does not fully meet PLOS ONE’s publication criteria as it currently stands. Therefore, we invite you to submit a revised version of the manuscript that addresses the points raised during the review process.

We look forward to receiving your revised manuscript.

Kind regards,

Pramod K Pandey

Academic Editor

PLOS ONE

Journal Requirements:

Additional Editor Comments (if provided):

Please see the comments from reviewer and attempt to resolve the issue. Overall, manuscript is improved, few issues are yet to be resolved. In addition, please revisit manuscript carefully to improve writing errors, and English of the manuscript.

Reviewers' comments:

Reviewer's Responses to Questions

**Comments to the Author**

1. If the authors have adequately addressed your comments raised in a previous round of review and you feel that this manuscript is now acceptable for publication, you may indicate that here to bypass the “Comments to the Author” section, enter your conflict of interest statement in the “Confidential to Editor” section, and submit your "Accept" recommendation.

Reviewer #1: All comments have been addressed

Reviewer #2: (No Response)

2. Is the manuscript technically sound, and do the data support the conclusions?

Reviewer #1: Yes

Reviewer #2: Yes

3. Has the statistical analysis been performed appropriately and rigorously? 

Reviewer #1: Yes

Reviewer #2: I Don't Know

4. Have the authors made all data underlying the findings in their manuscript fully available?

Reviewer #1: Yes

Reviewer #2: Yes

5. Is the manuscript presented in an intelligible fashion and written in standard English?

Reviewer #1: Yes

Reviewer #2: Yes

6. Review Comments to the Author

Reviewer #1: Dear Authors

Your response to the reviewer's comments is satisfactory.

Kind regards

Dr Masoud Shirali

Reviewer #2: The manuscript is well written; clear, precise, organized and easy to understand. With the minor revisions, I do recommend this manuscript for publication.

7. PLOS authors have the option to publish the peer review history of their article (what does this mean?). If published, this will include your full peer review and any attached files.

Reviewer #1: **Yes: **Dr Masoud Shirali

Reviewer #2: **Yes: **Mahmoud M. Nour, PhD

---

## [Author Response · Author response to Decision Letter 1]

14 Mar 2023

REVIWER’S SPECIFIC COMMENTS:

Reviewer #1 Dr Masoud Shirali:

We thank Dr Masoud Shirali for the positive comment.

Reviewer #2 Dr Mahmoud M. Nour: 

[Response]

Thank you very much for providing important comments. We are appreciate for the time and energy you expended. We revised as follows.

Line1 (Line1in the new file):

 After careful thought, we would like to change title from ‘Cattle-related injuries in Japan’ to ‘Cattle-related Occupational Accidents in Japan’. According to your kind advice, we made it more accountable about the study. 

Line23 (Line27 in the new file): 

 We would like to keep ‘them’ to describe more clearly. This ‘them’ means ‘extracted data set’ and it is different from the original data. 

Line24 (Line28 in the new file): 

 We would like to keep original sentence. Because we describe age as age group and not average age.

Line124 etc. (Line148 etc. in the new file): 

 We would like to keep original sentence. We would describe more pointed span and it is a set idiom.

Other pointed words and sentences:

 We revised according to your advice.

---

## [Decision Letter · Decision Letter 2]

17 May 2023

PONE-D-22-16745R2Cattle-related Occupational Accidents in Japan.PLOS ONE

Dear Dr. IWAI,

Thank you for submitting your manuscript to PLOS ONE. After careful consideration, we feel that it has merit but does not fully meet PLOS ONE’s publication criteria as it currently stands. Therefore, we invite you to submit a revised version of the manuscript that addresses the points raised during the review process.

We look forward to receiving your revised manuscript.

Kind regards,

Pramod K Pandey

Academic Editor

PLOS ONE

Journal Requirements:

Additional Editor Comments (if provided):

We have received sufficient feedback. Minor revisions are suggested. Authors should put additional efforts on improving the manuscript writing and discussion.

Reviewers' comments:

Reviewer's Responses to Questions

**Comments to the Author**

1. If the authors have adequately addressed your comments raised in a previous round of review and you feel that this manuscript is now acceptable for publication, you may indicate that here to bypass the “Comments to the Author” section, enter your conflict of interest statement in the “Confidential to Editor” section, and submit your "Accept" recommendation.

Reviewer #1: All comments have been addressed

Reviewer #3: All comments have been addressed

2. Is the manuscript technically sound, and do the data support the conclusions?

Reviewer #1: Yes

Reviewer #3: Yes

3. Has the statistical analysis been performed appropriately and rigorously? 

Reviewer #1: Yes

Reviewer #3: Yes

4. Have the authors made all data underlying the findings in their manuscript fully available?

Reviewer #1: Yes

Reviewer #3: Yes

5. Is the manuscript presented in an intelligible fashion and written in standard English?

Reviewer #1: Yes

Reviewer #3: Yes

6. Review Comments to the Author

Reviewer #1: (No Response)

Reviewer #3: Method

Wouldn't it be more useful to show Japan's agricultural population and the number of JA mutual aid subscribers rather than showing the number of JA subscribers?

Discussion

I recommend adding consideration to the following two points;

1) In the four years from 2013 to 2016, three work-related deaths caused by cattle have been confirmed.

https://www.jisha.or.jp/international/topics/pdf/2021j070101.pdf

2) In the following four years from 2017 to 2020, 207 such cases have been confirmed.

https://www.ja-kyosai.or.jp/files/2018/201808-3.pdf

Line 176: Isn't "dairy cattle" more common than "milk cattle"?

Lines 191 and 192: Ministry of Agriculture, Forestry and Fisheries

Line 193: result -> results

7. PLOS authors have the option to publish the peer review history of their article (what does this mean?). If published, this will include your full peer review and any attached files.

Reviewer #1: **Yes: **Dr Masoud Shirali

Reviewer #3: No

---

## [Author Response · Author response to Decision Letter 2]

29 Jun 2023

REVIWER’S SPECIFIC COMMENTS:

Reviewer #1: Dr Masoud Shirali

[Response]

Thank you very much for providing important comments. We appreciate for the time and energy you expended.

Reviewer #3: 

[Response]

Thank you very much for providing important comments. We appreciate for the time and energy you expended. We revised as follows.

1. The number of populations working for agriculture: Though, we didn’t write about it to keep our article simple, according to your advice, we added the number of populations working for agriculture (national population census of Japan in 2015). However, there are concerns that the number dose not fully reflect the actual situation. Because there so many part-time worker and family helper working in farm in busy farming season. And those people are not counted for population mainly working for agriculture. (Line 54~58)

2. The number of policyholders: Though, we didn’t write about it to keep our article simple, and JA Kyosai does not disclose the number of policyholders for individual insurance products, according to your advice, we added the released number of policyholders of comprehensive life insurance. We do not consider it important because our research dose not aim to calculate incident rate or other epidemiological indexes. (Line 89~93)

3. Additional consideration in discussion: We thank you for your deep interest in our work and your empathic advice. We carefully discussed your 2 points of suggestion. First your suggestion is about 3 fatal cases; the reference is the data of Survey on Industrial Incidents, of the Ministry of Health, Labour and Welfare of the Japanese government released by Japan Industrial Safety and Health Association. The data is from another survey; it is difficult to compare to our study directly. And our study discusses only injured case. Second your suggestion is about the report from JA Kyosai report in data of 2017 to 2020; This data represents a newer time period for the data we have been provided with and analysed. These data also need to be considered, but they do not influence the results of our study.　And as a consequence, we have decided to refrain from mentioning these subjects.

4. Thank you for your advice. We agree that ‘Milk cattle’ is more common as spoken. However, we would like to keep ‘Dairy cattle’ because it is more formal and academic. 

5. Thank you for correcting our miss typing. We correct it to ‘Ministry of Agriculture, Forestry and Fisheries’ and ‘results’.

---

## [Editor Report · Decision Letter 3]

14 Jul 2023

Cattle-related Occupational Accidents in Japan.

PONE-D-22-16745R3

Dear Dr. IWAI,

We’re pleased to inform you that your manuscript has been judged scientifically suitable for publication and will be formally accepted for publication once it meets all outstanding technical requirements.

Kind regards,

Pramod K Pandey

Academic Editor

PLOS ONE

Additional Editor Comments (optional):

Manuscript is acceptable. Figures quality needs to be improved during next stage of publication process.
---

## [Editor Report · Acceptance letter]

21 Jul 2023

PONE-D-22-16745R3 

Cattle-related Occupational Accidents in Japan. 

Dear Dr. Iwai:

I'm pleased to inform you that your manuscript has been deemed suitable for publication in PLOS ONE. Congratulations! Your manuscript is now with our production department. 

Kind regards, 

on behalf of

Dr. Pramod K Pandey 

Academic Editor

PLOS ONE